# First- and Second-Generation EGFR-TKIs Are All Replaced to Osimertinib in Chemo-Naive *EGFR* Mutation-Positive Non-Small Cell Lung Cancer?

**DOI:** 10.3390/ijms20010146

**Published:** 2019-01-03

**Authors:** Masayuki Takeda, Kazuhiko Nakagawa

**Affiliations:** Department of Medical Oncology, Kindai University Faculty of Medicine, 377-2 Ohno-higashi, Osaka-Sayama, Osaka 589-8511, Japan; nakagawa@med.kindai.ac.jp

**Keywords:** epidermal growth factor receptor (EGFR), tyrosine kinase inhibitor (TKI), mutation, non–small cell lung cancer (NSCLC), drug resistance

## Abstract

Activating mutations of the epidermal growth factor receptor gene (*EGFR*) are a driving force for some lung adenocarcinomas. Several randomized phase III studies have revealed that treatment with first- or second-generation EGFR tyrosine kinase inhibitors (TKIs) results in an improved progression-free survival (PFS) compared to standard chemotherapy in chemonaive patients with advanced non–small cell lung cancer (NSCLC), selected based on the presence of *EGFR* mutations. Patients treated with second-generation EGFR-TKIs have also shown an improved PFS relative to those treated with first-generation EGRF-TKIs. Osimertinib is a third-generation EGFR-TKI that still irreversibly inhibits the activity of EGFR after it has acquired the secondary T790M mutation that confers resistance to first- and second-generation drugs. Its efficacy has been validated for patients whose tumors have developed T790M-mediated resistance, as well as for first-line treatment of those patients with *EGFR* mutation–positive NSCLC. Although there are five EGFR-TKIs (gefitinib, erlotinib, afatinib, dacomitinib, and osimertinib) currently available for the treatment of *EGFR*-mutated lung cancer, the optimal sequence for administration of these drugs remains to be determined. In this review, we addressed this issue with regard to maximizing the duration of the EGFR-TKI treatment.

## 1. Introduction

Rapid developments in molecular biology provide the evidence that driver mutation, such as epidermal growth factor receptor (EGFR) [1,2] and anaplastic lymphoma kinase (ALK) genes [3], play an important role in the oncogenesis of non–small cell lung cancer (NSCLC). Indeed, randomized phase III studies revealed that first-line treatment with EGFR tyrosine kinase inhibitors (TKIs) conferred an improved progression-free survival (PFS) compared with standard chemotherapy in patients with advanced NSCLC, who were selected on the basis of the presence of activating *EGFR* mutations [4,5,6,7,8,9] (Table 1). Therefore, EGFR-TKI monotherapy has become the standard of care for patients with advanced NSCLC that are positive for such mutations. However, although most NSCLC patients who harbor TKI-sensitizing *EGFR* mutations show an initial pronounced response to EGFR-TKI treatment, they acquire a resistance to these drugs after ~9 to 14 months of such therapy.

Several mechanisms of acquired resistance to EGFR-TKIs—including the T790M secondary mutation in exon 20 of *EGFR*, *MET* amplification, overexpression of hepatocyte growth factor (HGF), and activation of the insulin-like growth factor 1 receptor (IGF1R)—have been identified [15,16,17,18]. The T790M mutation of *EGFR* is the most common mechanism of such an acquired resistance, having been detected in up to 50% of patients treated with the first-generation EGFR-TKIs erlotinib or gefitinib. Recent data indicates a similar frequency of T790M-mediated resistance in patients receiving first-line treatment with the second-generation EGFR-TKI afatinib [19]. The third-generation EGFR-TKI osimertinib was developed to overcome T790M-mediated acquired resistance to EGFR-TKIs, with this drug being an irreversible inhibitor of EGFR positive for T790M, but having little inhibitory activity for wild-type EGFR [20]. The efficacy of osimertinib has been validated in a phase III study (AURA3) that compared osimertinib with platinum-based doublet chemotherapy in advanced NSCLC patients that were positive for the T790M mutation of *EGFR* and whose tumors had progressed during previous EGFR-TKI therapy [13]. On the basis of these findings, osimertinib was assessed as a first-line treatment for *EGFR* mutation–positive NSCLC in comparison to a first-generation EGFR TKI (gefitinib or erlotinib) in the FLAURA trial, which demonstrated a significant improvement in PFS with osimertinib [14]. Given that *EGFR* mutation–positive tumors are highly dependent on EGFR signaling, a phenomenon referred to as “oncogene addiction”, the optimization of the sequence of administration of the five currently available EGFR-TKIs (erlotinib, gefitinib, afatinib, dacomitinib, and osimertinib) in patients with such tumors is warranted. This study addresses the optimal sequential therapy for EGFR-TKIs, with regard to maximization of the duration of the EGFR-TKI treatment in patients with *EGFR* mutation–positive NSCLC. We do not address the trials of EGFR-TKIs in combination with cytotoxic chemotherapy, such as platinum-doublet therapy, in order to focus on the therapeutic effects of the specific targeting of EGFR signaling pathways.

## 2. Comparison between the First-Generation EGFR-TKIs: Erlotinib versus Gefitinib (WJOG 5108L Trial)

Given that previous studies had focused on the assessment of the efficacy of first-generation EGFR-TKIs in comparison with platinum-doublet therapy in *EGFR*-mutated NSCLC, a multicenter, randomized phase III trial (WJOG 5108L) was designed to directly compare erlotinib with gefitinib for the treatment of advanced lung adenocarcinoma, regardless of the *EGFR* mutation status [21]. In December 2011, the protocol was amended to include only *EGFR* mutation–positive patients, given that the Pharmaceuticals and Medical Devices Agency (PMDA) of Japan decided that there was no indication for gefitinib in patients who were negative for the *EGFR* mutation. Among 561 patients enrolled, 198 (70.7%) and 203 (72.8%) *EGFR* mutation-positive patients were assigned to the erlotinib and gefitinib arms, respectively. Among the *EGFR* mutated NSCLC, the median PFS was 8.3 and 10.0 months for gefitinib and erlotinib, respectively (*p* = 0.424). Therefore, this study did not demonstrate non-inferiority of gefitinib compared to erlotinib in terms of PFS in patients with lung adenocarcinoma, according to the predefined criteria. However, the Kaplan–Meier survival for the two arms was almost identical, and these two first-generation EGFR-TKIs were considered almost equivalent in clinical practice.

## 3. Comparison between the First- and Second-Generation EGFR-TKIs: Gefitinib versus Afatinib (LUX-Lung 7) or Dacomitinib (ARCHER 1050)

Afatinib has a higher affinity for the kinase domain of EGFR compared with the first-generation EGFR-TKIs. The consequent irreversible blockade of tyrosine kinase activity might be expected to result in a more persistent suppression of EGFR signaling relative to the reversible inhibition achieved with erlotinib or gefitinib [22]. Given that the broader spectrum of activity and irreversible mechanism of action of afatinib was predicted to result in improved inhibition of EGFR-dependent tumor growth, compared with the first-generation EGFR-TKIs, a randomized, open-label phase IIb trial (LUX-Lung 7) of afatinib versus gefitinib was performed for the first-line treatment of patients with advanced lung adenocarcinoma who were positive for activating mutations (exon-19 deletions or the L858R point mutation) of *EGFR* [10]. The primary end points of the study were PFS, OS, and time to treatment failure. A total of 571 patients were screened, 319 of whom were randomized to the afatinib (*n* = 160) or gefitinib (*n* = 159) arms. Afatinib treatment was associated with a significantly improved PFS (median of 11.0 versus 10.9 months; HR = 0.73, *p* = 0.017) and time to treatment failure (median of 13.7 versus 11.5 months; HR = 0.73, *p* = 0.0073) compared with gefitinib.

Dacomitinib is a potent, second-generation EGFR-TKI that irreversibly binds EGFR, as well as the related proteins ErbB2 and ErbB4 [23]. Given the encouraging results of a phase II study of dacomitinib in the first-line setting [24], ARCHER 1050, a randomized, open-label phase III study of dacomitinib versus gefitinib, was conducted in treatment-naive patients with *EGFR* mutation–positive advanced NSCLC. In contrast to LUX-Lung 7, the ARCHER 1050 trial excluded patients with brain metastases. The primary end point of ARCHER 1050 was PFS, as assessed by a masked independent review of the intention-to-treat population. The results showed a statistically significant improvement in PFS in the dacomitinib cohort, with a median value of 14.7 months compared to 9.2 months in the gefitinib arm (HR = 0·59, *p* < 0.0001) [11]. The mature OS analysis for the intention-to-treat population was also recently published, with the results showing that dacomitinib treatment also led to a significant improvement in OS, with a median of 34.1 months compared to 26.8 months for gefitinib (HR = 0.76, *p* = 0.044) [25]. Thus, these two randomized studies suggested that second-generation EGFR-TKIs were superior to first-generation EGFR-TKIs, at least in terms of PFS.

## 4. Antiangiogenic Agents that Target the VEGF Pathway in Combination with First-Generation EGFR-TKIs

The vascular endothelial growth factor (VEGF) is a key regulator of tumor angiogenesis and likely contributes to the pathogenesis and progression of NSCLC. Given that antiangiogenic agents that target VEGF signaling show clinical activity for NSCLC when administered in addition to chemotherapy [26,27], a phase II study (JO25567) was undertaken in Japan to compare erlotinib alone with erlotinib plus bevacizumab, a monoclonal antibody to VEGF-A, as a first-line therapy in patients with advanced non-squamous NSCLC harboring *EGFR* mutations. This study found that PFS as a primary end point was longer with the combination treatment than with the erlotinib monotherapy (HR = 0.54, *p* = 0.0015) [28]. A subsequent confirmatory phase III study that compared erlotinib plus bevacizumab, with erlotinib alone, in patients with untreated NSCLC that were positive for activating *EGFR* mutations (NEJ 026) showed that the median PFS, as determined by independent review, was 16.9 months compared to 13.3 months (HR = 0.605, *p* = 0.016), respectively [12] (Table 1). Final data for OS, a secondary end point, are not yet available. Therefore, bevacizumab plus erlotinib has become a new standard therapy for treatment-naive patients with *EGFR* mutation–positive NSCLC.

Ramucirumab is a monoclonal immunoglobulin G1 antibody that binds to the extracellular domain of the VEGF receptor VEGFR-2 with high specificity. Given that several trials have found that bevacizumab in combination with an EGFR-TKI might provide additional clinical benefits in NSCLC patients with activating *EGFR* mutations [28,29], a randomized phase Ib/III study (RELAY) to investigate the safety and efficacy of the combined use of ramucirumab and erlotinib in the first-line setting for patients with stage IV NSCLC positive for *EGFR* mutations is also now underway [30].

## 5. Comparison of a Third-Generation EGFR-TKI with Platinum-Doublet Chemotherapy in NSCLC Positive for *EGFR* T790M

Osimertinib, an irreversible T790M mutant–specific EGFR-TKI with little inhibitory activity for wild-type EGFR, was developed to overcome T790M-mediated acquired resistance to first- or second-generation EGFR-TKIs [20]. AURA3, an open-label, randomized phase III study, was performed to assess the efficacy and safety of osimertinib versus platinum-based doublet chemotherapy in patients with advanced NSCLC that were positive for the T790M mutation of *EGFR* that had progressed during previous EGFR-TKI therapy [13]. The primary end point of the trial was PFS, and the secondary end points included OS, overall response rate, duration of response, disease control rate, safety, and measures of health-related quality of life. Osimertinib conferred a statistically significant improvement in PFS compared to standard platinum-doublet chemotherapy (median of 10.1 versus 4.4 months; HR = 0.30, *p* < 0.001) (Table 1). PFS was also significantly longer with osimertinib than with platinum-based doublet chemotherapy (8.5 versus 4.2 months; HR = 0.32, with a 95% confidence interval of 0.21–0.49) in the 34% of patients with central nervous system (CNS) metastases at the baseline. In November 2015, the U.S. Food and Drug Administration (FDA) approved osimertinib in the form of 80-mg once-daily tablets for the treatment of patients with metastatic NSCLC positive for *EGFR* T790M (as detected with an FDA-approved test), who had progressed during or after prior EGFR-TKI therapy.

## 6. Comparison of the First- and Third-Generation EGFR-TKIs: Erlotinib or Gefitinib versus Osimertinib (FLAURA Trial)

Given the encouraging results of the AURA trial of osimertinib (administered at 80 or 160 mg daily) as a first-line treatment for patients with *EGFR* mutation–positive advanced NSCLC, which revealed a median PFS of 20.5 months [31], osimertinib has been evaluated in a randomized phase III trial (FLAURA) in comparison to a standard first-generation EGFR-TKI (gefitinib at 250 mg daily, or erlotinib at 150 mg daily), for treatment-naive patients with *EGFR*-mutated metastatic NSCLC in the first-line setting. The results to date have shown that osimertinib was associated with a longer PFS compared to the standard of care (median of 18.9 versus 10.2 months; HR = 0.46, *p* < 0.0001) [14] (Table 1). This benefit was maintained across all the prespecified subgroups, including patients with CNS metastases at study entry. The median PFS values were 15.2 and 9.6 months (HR = 0.47, *p* = 0.0009), respectively, for the latter patients, whereas they were 19.1 and 10.9 months (HR = 0.46, *p* < 0.0001) for those patients without CNS metastases. The objective response rate was similar between the two treatment groups (80% with osimertinib versus 76% with the standard of care). Although the data were immature at the time of the interim analysis, there was a trend toward improved OS with osimertinib that had not yet reached statistical significance. Based on these promising results, osimertinib obtained an additional FDA indication for the first-line treatment of patients with metastatic NSCLC positive for exon-19 deletions or the L858R point mutation of *EGFR* (again as detected by an FDA-approved test).

## 7. What Is the Best EGFR-TKI Sequence for Treatment?

Regardless of which first- or second-generation EGFR-TKI is selected, the development of resistance is inevitable, usually around 9 to 14 months after the treatment onset. Although several mechanisms of resistance to these drugs have been identified, the mechanisms of resistance to osimertinib in the first-line setting have not been fully elucidated. A recent retrospective analysis of the FLAURA trial looking at the mechanisms of acquired resistance to first-line osimertinib in *EGFR*-mutated advanced NSCLC, found that the C797S mutation of *EGFR*, which also confers osimertinib resistance, was present at a low frequency (7%) in patients who had acquired a resistance to osimertinib. The most common acquired resistance mechanism detected was *MET* amplification (15%), followed by *PIK3CA* (7%) and *KRAS* (3%) mutations and *HER2* amplification (2%) [32]. Such findings weaken support for a treatment strategy of osimertinib followed by other EGFR-TKIs, given the lack of targeted treatment options after osimertinib failure, with chemotherapy being the most prevalent therapeutic choice. Indeed, of the 59% of patients who received therapy after osimertinib in the FLAURA trial, 56% received chemotherapy and 35% received an EGFR-TKI–based regimen [33].

Treatment options in real-world clinical practice were evaluated in a retrospective observational study for TKI-naive patients with advanced NSCLC who were treated in the first-line setting with afatinib, and acquired the T790M mutation of *EGFR*, and then received osimertinib. Although inclusion was restricted to patients who initiated osimertinib treatment at ≥10 months before enrollment in order to avoid early censoring and to ensure data maturity, the median time on treatment for sequential afatinib and osimertinib was 27.6 months [34]. Therefore, a first- or second-generation EGFR-TKI followed by osimertinib may become a standard treatment option for chemotherapy-naive patients with *EGFR*-mutated NSCLC. This approach would be supported by the ability to identify patients likely to develop T790M from the analysis of a pretreatment tissue sample. However, most physicians hesitate to administer a first- or second-generation EGFR-TKI in the first-line setting, given that the indication for osimertinib in the second-line setting is limited to patients with metastatic NSCLC, positive for *EGFR* T790M (as detected by an FDA-approved test), whose disease has progressed during or after prior EGFR-TKI therapy.

There is currently no clear evidence to support the selection of patients at diagnosis, who are likely to develop T790M after treatment with a first- or second-generation EGFR-TKI. Tumor mutation burden (TMB) in pre–EGFR-TKI tumor specimens is a potential biomarker for the prediction of T790M-mediated resistance. A recent study assessed the impact of TMB on the outcome of patients with *EGFR*-mutated NSCLC [35]. Among patients who underwent *EGFR* T790M testing at the time of the development of resistance to first- or second-generation EGFR-TKIs, the median TMB for the pre–EGFR-TKI sample of those patients that acquired T790M at resistance was 3.77 mutations/Mb, versus a value of 4.72 mutations/Mb for those patients who did not. These results suggest that it may be possible to select patients that are likely to develop the T790M mutation on the basis of TMB if an optimal cut-off value can be determined (Figure 1).

In conclusion, although several EGFR-TKIs are now available in clinical practice, the best sequence for administration of these drugs with regard to maximization of the duration of the EGFR signaling inhibition has not been determined. Comprehensive characterization of resistance mechanisms for each EGFR-TKI will contribute to the development of more effective strategies.

## Figures and Tables

**Figure 1 ijms-20-00146-f001:**
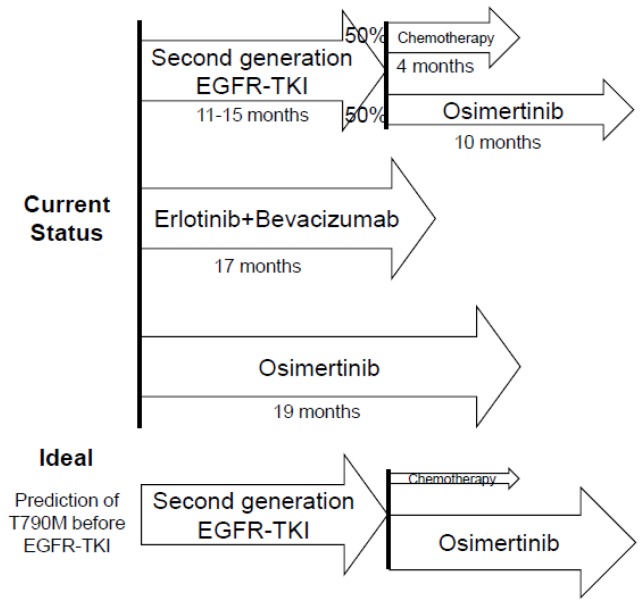
Current and potential ideal sequential treatment regimens with EGFR-TKIs in patients with advanced NSCLC harboring *EGFR* mutations. Median PFS values are indicated.

**Table 1 ijms-20-00146-t001:** Median progression-free survival (PFS) in clinical trials for patients with *EGFR* mutation–positive advanced non–small cell lung cancer (NSCLC) treated with EGFR-TKIs.

Regimen	Trials	Median PFS (Months)	References
Gefitinib	WJTOG3405, NEJ002, LUX-Lung 7, ARCHER 1050	9.2–10.9	[4,5,10,11]
Erlotinib	EURTAC, OPTIMAL, NEJ026	10.4–13.3	[6,7,12]
Afatinib	LUX-Lung 3, LUX-Lung 6, LUX-Lung 7	11.0–11.1	[8,9,10]
Dacomitinib	ARCHER 1050	14.7	[11]
Erlotinib + Bevacizumab	NEJ026	16.9	[12]
Osimertinib (second line)	AURA3	10.1	[13]
Osimertinib (first line)	FLAURA	18.9	[14]

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
