# Peer review of "First- and Second-Generation EGFR-TKIs Are All Replaced to Osimertinib in Chemo-Naive EGFR Mutation-Positive Non-Small Cell Lung Cancer?"

_ijms, 2019, doi:10.3390/ijms20010146_

Round 1

Reviewer 1 Report

The manuscript clearly compared osimertinib with the first and second generation TKIs. The evidence and clinical trials mentioned in the article are clear to support the authors' concept. This is a good review paper with well organization to summarize 3 generations of EGFR-TKI. 

However, the comparisons between these EGFR-TKI have been performed in some reports, and therapeutic effect of osimertinib on resistant NSCLC has been not news.

Although this manuscript dose not provide a novel concept, it still provide a good summary to illustrate the progression of EGFR-TKI development. This paper will help lung cancer researcher to realize the progression of target therapy in NSCLC. Hence, I suggest IJMS to accept this manuscript for publication.

Author Response

Thank you for reviewer’s positive comments. As pointed out by reviewer, osimertinib have been established as first-line treatment for chemo-naive EGFR mutated NSCLC; however, EGFR-TKI sequence such as afatinib followed by osimertinib still be possible treatment strategy, if emergence of T790M can be predicted in pre-EGFR-TKI tissues. 

Reviewer 2 Report

In the review article entitled 'Osimertinib Versus First- or Second-Generation EGFR TKIs for the Treatment of Chemotherapy Naive Patients with EGFR Mutant-Positive Non-Small Cell Lung Cancer', the authors compared different clinical trials using five EGFR-TKIs. The authors also discussed the sequence of EGFR TKIs for treatment of mEGFR NSCLC. However, the reviewer did not feel the current manuscript is acceptable for publication. The major concerns are listed below:

Lack of clinical reference. On April 18, 2018, osimertinib received FDA approval for first-line treatment of patients with metastatic EGFR-mutant NSCLC. The most recent guidelines from the NCCN also recommend the use of first-line osimertinib. These notable changes represent a paradigm shift in the treatment of patients with EGFR-mutant NSCLC. As a result, T790M will not be an important factor to consider since the EGFR-mutant patients will soon received first-line osimertinib. Therefore, the review feel it not clinically relevant to still focus on T790M and 1st/2nd generation of EGFR TKIs.

The title is misleading. The title indicated the manuscript will focus on the difference between osimertinib vs. 1st/2nd generations of EGFR TKIs. However only section 6 in the manuscript are on this topic. Suggest the authors choose a more suitable title for the manuscript.

Author Response

We thank the reviewer for insightful comments, which we feel have helped us to improve our manuscript.

We almost agree to reviewer’s comments that osimertinib have been established as first-line treatment for chemo-naive EGFR mutated NSCLC. Strategy of osimertinib followed by the other EGFR-TKI may be limited in EGFR mutated NSCLC, because the mechanisms of acquired resistance to first-line osimertinib in advanced EGFR mutated NSCLC have shown that C797S mutation associated with EGFR-TKI sensitivity occurred at low frequency (7%). Therefore, EGFR-TKI sequence such as afatinib followed by osimertinib still be possible treatment strategy, if emergence of T790M can be predicted in pre-EGFR-TKI tissues.

Accordingly, we have changed the title as "First and second generation EGFR-TKIs are all replaced to osimertinib in chemo-naive EGFR mutation-positive non-small cell lung cancer?".

Round 2

Reviewer 2 Report

Thank you for the detailed reply and further revision. I think the revised title represents the scope of this manuscript better.